# The impacts on food purchases and tax revenues of a tax based on Chile's nutrient profiling model

**M. Arantxa Colchero**[1], **Guillermo Paraje**[2], **Barry M. Popkin**[3]*

**1** Instituto Nacional de Salud Pública, Morelos, México, **2** Escuela de Negocios, Universidad Adolfo Ibáñez, Peñalolén, Chile, **3** Department of Nutrition and Carolina Population Center, The University of North Carolina at Chapel Hill, North Carolina, United States of America

* popkin@unc.edu

## Abstract

### Background

In June 2016, Chile implemented the Law of Food Labelling and Advertising, which included a mandatory front-of-pack warning labels on food and beverages high in added sugar, saturated fat, sodium or energy density, restrictions on child-directed marketing and on the promotion and sales of these products in schools. The regulation does not include taxes although Chile had implemented a tiered tax on SSBs two years before this law was implemented. Therefore, the objective of the study was to simulate the impact of taxing food and beverages based on the cutoff's points for warning labels on purchases and revenues.

### Methods

We derived price elasticities using the linear approximation of the almost ideal demand system for six groups of labeled food and beverages (with a warning label based on the regulation) and unlabeled (with no warning label): 1) unlabeled beverages, 2) labeled beverages, 3) unlabeled cereal based products, 4) labeled cereal based products, 5) labeled meat and fish and 6) labeled sweet snacks and desserts. The study used data on household food beverage purchases from the Kantar WorldPanel Chile and Euromonitor sales to adjust the Kantar elasticity results to the national average. We estimated revenues under three tax scenarios for all labeled food and beverages: 10%, 20%, 30% of the final price excluding taxes.

### Results

Except for labeled fish and meat, all food and beverage groups were price elastic. After accounting for a reduction in consumption after the taxes, economic and population growth, revenues for all groups could reach between 457 million USD to 1.3 billion USD. These results based on the much larger tax base of these labeled "high in added sugar, salt or saturated fat or energy density" foods and beverages is much larger.

**Data Availability Statement:** The data supporting the results of this study are owned by a private company. https://www.kantarworldpanel.com/cl/Sobre-nosotros Av. del Valle Nte. 928, Huechuraba, and the info of our contact person is: María Paz

Román email address: mariapaz.roman@kantar.com The Euromonitor data can be found here https://go.unc.edu/gfrp-data-euromonitor-chile.

**Funding:** Funding support comes from Bloomberg Philanthropies (www.bloomberg.org) with support also from the Carolina Population Center NIH Center grant (P2C HD0550924) The funders had no role in study design, data collection and analysis, decision to publish, or preparation of the manuscript. The authors maintained independence from funders and all authors had access to statistical reports and tables of the study and can take responsibility for the integrity and accuracy of the data. The funders had no role in study design, data collection and analysis, decision to publish, or preparation of the manuscript.

**Competing interests:** The authors have declared that no competing interests exist.

## Conclusion

This fiscal package could be implemented in countries with warning labels to enhance health and welfare. The Chilean warning label front-of-the-package system provides an important guide for countries considering policies to reduce diet-related non communicable diseases, including obesity. The fiscal policy impact alone, as shown here for Chile, will be highly impactful in reducing ultra-processed food intake and generating revenues.

## Introduction

Fiscal policy has been utilized globally focused on sugar-sweetened beverages (SSBs) [1–3] or limited categories of non-essential foods [4–6] increasingly now called ultra-processed food [7]. Ultra-processed food and SSB are industrialized formulations made of substances derived from food or synthesized from other organic sources, combined with various additives, which require little or no culinary preparation for consumption [8]. To date, no one has created a tax that focused uniformly across all categories of foods and beverages that were deemed unhealthy. Chile used a nutrient profile model that identified foods high in added sugar, saturated fat, or sodium and energy density to create a comprehensive set of laws. In June 2016, Chile implemented the Law of Food Labeling and Advertising, which included the first national system of mandatory front-of-pack (FOP) warning labels for SSBs and energy-dense, non-essential foods [9, 10]. Chile's Law of Food Labeling and Advertising also exacts comprehensive restrictions on child-directed marketing of SSBs and non-essential, energy-dense foods to children under 14 years of age, as well as restrictions on the promotion and sales of these products in schools. The regulation was designed to be implemented in three phases with increasing nutrient thresholds. Phase 1 was implemented in June 2016, phase 2 in June 2018 and phase three in 2019 with increasingly stringent thresholds for all nutrients and energy density [11].

This comprehensive regulation does not include taxes although Chile has an earlier tax on SSBs but not on non-essential energy dense food. Chile had a 13% tax applied to SSB that was modified in October 2014 to a tiered tax. The tax rate increased to 18% for non-alcoholic beverages, naturally or artificially flavored with high levels of sugar (greater than 6.25 grams of sugar per 100 milliliters) and decreased to 10% to beverages with less than 6.25 grams per 100ml.

There is a large literature on the role of sin taxes for reducing consumption and improving health and welfare [12, 13]. However in the food area most of this focus has been on SSB taxation and over 42 countries and a number of cities and other jurisdictions have instituted such policies [2, 14]. These taxes typically stand alone and are not used to reinforce other related laws. In contrast, Chile, along with Israel, Mexico, Peru, and Uruguay, either have or will soon have (i.e. Mexico passed a law but is in the regulation phase) nutrient profiling models used for front-of-the-package warning labels based on Chile's approach. This means the identified foods contribute not only to obesity but also the most prevalent noncommunicable diseases (NCD) such as diabetes and hypertension. A tax based on such a set of laws will not only reinforce the current law in Chile but add an additional push across a broader array of foods and beverages identified to be most unhealthy.

Ultra-processed foods have been shown a direct impact on weight gain. A random controlled trial run by the US National institutes of Health showed that a diet of ultra-processed food led to a weight gain of 0.9kg over 2 weeks compared with 2 weeks of a real food-based

diet [15]. This NIH work was amplified by several papers that came out two weeks later in the *British Medical Journal* that looked at two large European cohorts and showed a strong positive relation between ultra-processed foods and cardiovascular disease and all-cause mortality [16–19]. A large number of studies published earlier reported longitudinal data from children and adults that associated ultra-processed food intake with increased NCD risk [17–32]. Despite this work and the aforementioned existing evidence on the effectiveness of sin taxes to decrease consumption of unhealthy products, no country to date has used fiscal policy to promote reduction in consumption of these foods and beverages. Chile's first year of a set of increasing stringent guidelines has already produced a significant decline in SSB intake in a country with the highest per capita SSB sales in the world in 2014 [33]. The objective of the study was to simulate the impact of taxing food and beverages based on the Chilean food and beverage cutoffs points for warning labels [9] on purchases and revenues that could be used for other programs and policies. Estimates of fiscal revenues in this study are representative of food and beverages labeled in the third phase of the regulation.

## Methods

### Participants and data base

The University of North Carolina's Institutional Review Board approved of this study. All heads of households > 18 years age interviewed by Kantar have provided written approval. We used the Kantar WorldPanel Chile, a longitudinal household food and beverage purchase data from January 1, 2015 to December, 2017. The response rate for participation in Kantar World-Panel Chile is 95%. The Kantar WorldPanel is an open cohort, it follows households over time but can include new households. Data include household purchases of consumer-packaged goods from a panel of households located in cities with >20,000 inhabitants, representative of Chile's urban population. With replacement, our analytic sample had 2,383 unique households, with average follow-up of 29·2 months, providing 69,696 household-month observations. Enumerators visited households weekly to collect data on food and beverage purchases. Information on each purchase was collected either by scanning product barcodes using a handheld barcode scanner or by using a codebook to assign barcodes for bulk products or other products without barcodes. Interviewers also reviewed weekly receipts, conducted household pantry inventories, and checked empty product packages stored in a bin between interviews to ensure products were not double-counted. Data collected on each purchase included volume or weight, bar code, price per unit, retail channel, brand, package size, and date of purchase. Data were analyzed at the household-monthly level. Contact for obtaining the Kantar data can be obtained from the corresponding author.

An alternative to using Kantar and Euromonitor data is noted here. Euromonitor provides sales data for the country for all key foods. The data can be organized into the same categories as our simulation and provides total sales. You can use the National Income and Expenditures data for Chile named the Family Budget Survey (FBS) [34] to calculate elasticities using prices derived from expenditures and quantities. The expenditure survey data results would be linked to total Euromonitor sales to expand the data to include total food sales for each category. From the FBS, we can derive unit values (prices), using quantity purchased and expenditures to get the price structure. The annual national consumption to get revenues can be estimated by using purchases from the FBS and a correction factor using Euromonitor -given the potential underestimation of purchases using household expenditure data-. The expected annual consumption after the tax can be estimated using the income and own and cross price elasticities derived from the FBS and the price structure.

## Nutrient facts panel data and categorization by regulation status

We obtained nutrition facts panel (NFP) data from product photographs collected by a team of Chilean nutrition research assistants in stores during the first quarters of 2015, 2016, and 2017. We then linked NFP data at the product level to household food and beverage purchases using a similar process as in previous household purchase evaluations [1, 35, 36]. For the pre-regulation period, we linked purchases to NFP data collected in 2015 and 2016 (i.e., data reflecting the nutritional profiles of products available prior to the regulation). For the post-regulation period, we linked purchases to NFP data collected in 2017. If there was no direct 2017 link, we linked to the 2015–2016 NFP data. Linkages were based on barcode, brand name, and product description. Of total beverage purchases, 95·6% were linked to collected NFP data. If no collected NFP data was available for a purchased product, it was linked to Mintel Latin America (4·4%) a private company that collects data monthly for all new food and beverage products and their nutrient content, or other NFP data resources (<0·1%).

After linking the data, a team of Spanish-speaking nutritionist research assistants at the University of North Carolina and the University of Chile categorized each food and beverage purchase as to whether it should be subject to regulation according to the first phase nutrient profile model established by the Chilean regulation. Beverages and foods were categorized as "high-in" (and thus subject to regulation) if they contained added sugar, added sodium, or added saturated fat and exceeded the nutrient thresholds set in the first phase of implementation (i.e., >100 calories, >100mg sodium, >6g sugar, or 3g saturated fat per 100mL of product in its as-consumed form). These "high in" items would be required to carry a FOP warning label and be subject to the regulation's marketing and school sales restrictions. Food and beverages were considered "not high-in" if they did not meet these nutritional criteria. Food and beverages that were categorized as high-in were classified as such because they contained added sugar and their total sugar content exceeded 6g/100mL. We also classified foods and beverages into subgroups, including unlabeled and labeled (i.e. high-in) beverages, unlabeled cereal products, labeled meat and fish products, and labeled sweet snacks and desserts.

## Statistical analyses

**Estimation of price elasticities.** We estimated price elasticities using a demand system for food and beverages called the Linear Approximation of the Almost Ideal Demand System (AIDS) developed by Deaton and Muelbauer [37]. The model consists of a system of equations, one equation for each food or beverage group. In each equation, the dependent variable is the beverage and food expenditure share as a function of prices, total expenditures on food and beverages and the Fisher price. For each food and beverage group, we derived own price elasticities (how purchases of a specific food or beverage group changes with changes in prices of this same group) and cross price elasticities (how purchases of a specific group change with changes in prices of the other groups). We also modeled the quadratic AIDS (QUAIDS) model [38] in sensitivity analysis to see how robust the results were. The AIDS and QUAIDS models account for households with zero purchases. The model takes average unit values as proxy for prices for all households (purchasing or goods or not) and considers households with a zero-budget share [39]. Restrictions on homogeneity and symmetry were imposed in the AIDS and QUAIDS models. We derived confidence intervals for price and income elasticities using a bootstrap program with 300 replications.

We classified food and beverages into six categories based on the FOP warning label regulation: 1) unlabeled beverages (plain water, unlabeled milk, dairy, coffee and tea and 100% juice), 2) labeled beverages (sugar sweetened beverages, labeled milk, dairy, coffee and tea), 3) unlabeled cereal based products (grains, pasta, noodles, oats, crackers, bread, salty snacks), 4)

labeled cereal based products (ready to eat cereals, cookies, bread and rolls, salty snacks), 5) labeled meat and fish (fish, pork, poultry, cheese) and 6) labeled sweet snacks and desserts (candies, chocolate, desserts, labeled fruits). Two groups were excluded from the analyses: unlabeled fruit and beverages and unlabeled meat and fish that represented only 0·6% and 0·1% of total expenditures, respectively. We also excluded infant, toddler, maternal, and elderly formulas.

For each group, the dependent variable in the AIDS and QUAIDS models is the food or beverage expenditure share (proportion of household expenditures in each group over total expenditures on food and beverages in the system). We adjusted the models for year, quarter, household size and composition, household assets, education of the head of the household and we obtained robust standard errors using region as a cluster. We created a Fisher Price Index for each group that were included in the models. A price index summarizes prices of a basket of goods using a normalized value, a weighted mean of prices. A Fisher Price Index is a consumer price index that measures the price level of goods estimated as the geometric mean of the Laspeyres and Paasche index [40]. The Laspeyres index uses baseline prices calculated as the weighted sum of the baseline prices times consumption at time t over the weighted sum of baseline prices times consumption at baseline. The Paasche index is calculated as the weighted sum of prices multiplied by consumption at time t over prices at baseline multiplied by consumption at time t. In demand systems, Fisher Price Index reduce biases related to unit values [41–43]. To estimate each Fisher Price Index, we first derived brand-level unit values by dividing household purchases (in Chilean pesos) of products from each brand by the quantity purchased by the household at a specific quarter/year. For Laspeyres and Passche Index, the baseline value should be the first month or quarter of 2015. However, because after baseline, during the study period new brands may have appeared, we decided to take the average of all years. As the data set had many brands, we aggregated brands of products with less than 1% of market share volume into a single composite brand. We ran regressions at the food and beverage group to impute brand-level unit values when missing, adjusting for education, age of the head of the household, tertiles of household assets, time, and unemployment rates. For each household, unit values were calculated at the brand-quarter-year level. Because households purchase each quarter products from some but not all existing brands, for each household we imputed brand prices not purchased in each quarter. From 2015–2017, considering all household-brand-quarter-year possible purchases, we imputed had 82% imputed unit values.

Analyses were conducted using Stata 15.

**Estimation of tax revenues.**   We estimated tax revenues under scenarios of 10%, 20% and 30% tax on labeled food and beverages in three steps. We applied the excise taxes for each scenario to the same base as the Value Added Tax (VAT). The VAT in Chile is 19%. Chile already has a tiered SSB tax of 18% for beverage with high sugar content and 10% for lower sugar content. For consistency with the scenarios simulated in our study for all food and beverages with a warning label we applied a 10%, 20% and 30% tax to SSB as if no taxes existed. We first derived the national consumption in Chilean pesos by multiplying the average per capita volume purchased (ml or grams/day) from Kantar for each group multiplied by 365 days in the year, the 2017 population [44] and the average price/ml or gram (unit values derived from Kantar). Because purchase data could be underestimated as in any household survey compared to consumption data and given that Kantar only represents urban areas, to get the national consumption, we applied a correction factor based on sales data from Euromonitor [45]. The correction factor was estimated for each food or beverage group in 2017 by dividing Euromonitor´s per capita/day sales by the average per capita/day volume purchased in Kantar. In a second step, we estimated the expected national consumption after the tax by applying the income and prices elasticities (own and cross price elasticities) derived from the AIDS model,

assuming a 100% pass through prices and considering the economic and population growth between 2017 and 2018. There is evidence from different countries that taxes fully passed to consumers, not only for SSB, but also for alcohol and tobacco. Although there is little evidence in Chile of pass through prices, a recent paper showed that the SSB tax rate increase implemented in 2014 was effective to increase price, for carbonates price increases exceeded the tax change [46]. Expected consumption would in general decrease after the tax (based on own price elasticities) but as taxes are applied to all labeled food and beverages, substitutions or complementariness are plausible and expected consumption after the tax for each group can change. Finally, to get revenues, we multiplied the expected national consumption by the percent that the tax represents to the final price assuming that in Chile taxes are applied to wholesale prices.

## Results

Table 1 shows the distribution of expenditures, percent purchasers and Fisher Price Index for each food and beverage group and year. Beverages represent about half of total expenditures in this food and beverage system. As shown in the table, non-purchases of any food or beverage

**Table 1. Distribution of expenditures, percent purchasers and Fisher Price Index by food and beverage group and calendar year, 2015–2017.**

|  | All years | 2015 | 2016 | 2017 |
|---|---|---|---|---|
| Distribution of expenditures |  |  |  |  |
| Unlabeled beverages | 26.3% | 23.7% | 26.2% | 29.1% |
| Labeled beverages | 25.5% | 29.2% | 25.2% | 21.9% |
| Unlabeled cereal based products | 15.2% | 15.6% | 15.1% | 14.8% |
| Labeled cereal based products | 14.4% | 12.2% | 15.1% | 15.8% |
| Labeled meat and fish | 11.3% | 11.9% | 11.2% | 11.0% |
| Labeled sweets and desserts | 7.3% | 7.4% | 7.2% | 7.4% |
| Percent purchasers (purchases >0) |  |  |  |  |
| Unlabeled beverages | 99.5% | 99.4% | 99.6% | 99.5% |
| Labeled beverages | 99.5% | 99.7% | 99.6% | 99.4% |
| Unlabeled cereal based products | 99.0% | 99.2% | 99.1% | 98.8% |
| Labeled cereal based products | 98.4% | 98.4% | 98.7% | 98.0% |
| Labeled meat and fish | 96.5% | 97.3% | 96.5% | 95.6% |
| Labeled sweets and desserts | 94.6% | 95.4% | 94.6% | 93.8% |
| Fisher price index |  |  |  |  |
| Unlabeled beverages | 0.871 | 0.871 | 0.872 | 0.872 |
| Labeled beverages | 0.958 | 0.959 | 0.959 | 0.958 |
| Unlabeled cereal based products | 0.951 | 0.947 | 0.950 | 0.955 |
| Labeled cereal based products | 0.981 | 0.985 | 0.978 | 0.980 |
| Labeled meat and fish | 0.963 | 0.966 | 0.962 | 0.962 |
| Labeled sweets and desserts | 0.982 | 0.982 | 0.982 | 0.983 |
| Fisher price index (2015 = 100) |  |  |  |  |
| Unlabeled beverages | .. | 100.00 | 100.14 | 100.16 |
| Labeled beverages | .. | 100.00 | 99.97 | 99.94 |
| Unlabeled cereal based products | .. | 100.00 | 100.31 | 100.88 |
| Labeled cereal based products | .. | 100.00 | 99.29 | 99.49 |
| Labeled meat and fish | .. | 100 | 99.64 | 99.59 |
| Labeled sweets and desserts | .. | 100 | 100.01 | 100.14 |

Own estimations using Kantar data from January 2015 to December 2017.

**Table 2. Own, cross price elasticities and expenditure elasticities for food and beverages 2015–2017 (AIDS model).**

| Group | Unlabeled beverages | Labeled beverages | Unlabeled cereal based products | Labeled cereal based products | Labeled fish and meat | Labeled sweet and desserts |
|---|---|---|---|---|---|---|
| Price elasticities | | | | | | |
| Unlabeled beverages | -1.049 | 0.084 | 0.131 | -0.153 | -0.004 | -0.081 |
| | [-1.049,-1.047] | [0.083,0.084] | [0.129,0.131] | [-0.153,-0.151] | [-0.005,-0.002] | [-0.083,-0.0790] |
| Labeled beverages | 0.076 | -1.092 | -0.008 | 0.061 | -0.195 | 0.244 |
| | [0.075–0.077] | [-1.095,-1.089] | [-0.010,-0.005] | [0.057,0.065] | [-1.198,-1.192] | [0.238,0.250] |
| Unlabeled cereal based products | 0.033 | -0.045 | -1.113 | 0.133 | -0.139 | 0.225 |
| | [0.032–0.034] | [-0.046,-0.043] | [-1.116,-1.109] | [0.130,0.135] | [-0.142,-0.135] | [0.211,0.240] |
| Labeled cereal based products | -0.082 | 0.038 | 0.166 | -1.196 | 0.171 | -0.058 |
| | [-0.084,-0.082] | [0.035,0.040] | [0.163,0.168] | [-1.199,-1.192] | [0.168,0.173] | [-0.067,-0.048] |
| Labeled fish and meat | -0.007 | -0.091 | -0.078 | 0.129 | -0.820 | -0.023 |
| | [-0.008,-0.007] | [-0.092,-0.089] | [-0.081,-0.0753] | [0.126,0.130] | [-0.823,-0.817] | [-0.027,-0.018] |
| Labeled sweet and desserts | -0.027 | 0.067 | 0.125 | -0.034 | -0.015 | -1.306 |
| | [-0.027,-0.026] | [0.065,0.069] | [0.118,0.132] | [-0.029,-0.029] | [-0.018,-0.0125] | [-1.306,-1.300] |
| Income elasticities | 1.057 | 1.040 | 0.776 | 1.060 | 1.003 | 0.999 |
| | [1.056,1.057] | [1.038,1.040] | [0.775,0.777] | [1.058,1.060] | [1.001,1.004] | [0.997,1.000] |

Own estimations using Kantar data from January 2015 to December 2017. Models adjusted for household size and composition, year, quarter. 95% confidence intervals in brackets derived from bootstrapping, 300 replications.

within each group is very small, the percent of purchases greater than zero is more than 95% for all groups.

Table 2 presents the own and cross price elasticities using the AIDS model with 95% confidence intervals obtained from bootstrapping. All coefficients were statistically significant. Except for labeled fish and meat, all food and beverage groups are price elastic. As described in the Methods, cross price elasticities were considered to estimate national expected consumption and revenues. If taxes are applied to all labeled food and beverages, substitutions are plausible and expected consumption after the tax for each group can change. For example, Table 2 shows that labeled beverages increase with an increase in prices of labeled cereal based products and labeled sweet snacks and desserts and decreases with an increase in prices of labeled meat and fish. These changes in labeled beverages are included in the estimation of expected consumption to estimate revenues as well as for the other labeled food groups. Results from QAIDS are very similar compared to the AIDS model (S1 Table).

Table 3 presents the estimation of revenues from three tax scenarios on labeled food and beverages. Revenues on expected consumption (considering reductions based on price elasticities, changes from population and economic growth as well as income elasticities), sum

**Table 3. Revenue estimations for tax scenarios on food and beverages (millions of 2019 Chilean pesos).**

| Group | 10% tax | 20% tax | 30% tax |
|---|---|---|---|
| Labeled beverages | 194,244.1 | 371,816.9 | 562,107.4 |
| Labeled cereal based products | 50,414.9 | 83,828.3 | 156,441.0 |
| Labeled fish and meat | 62,081.0 | 117,381.6 | 165,174.8 |
| Labeled sweet and desserts | 81,852.0 | 156,598.7 | 220,593.7 |
| Total | 388,591.9 | 729,625.5 | 1,104,316.8 |

Own estimations using Kantar data from January 2015 to December 2017 and other parameters listed in S2 Table.

388,591.9 million pesos with a 10% tax, 729,625.5 with a 20% tax and 1,104,316.8 with a 30% tax. As the tax is applied to the same base as the VAT for all food and beverages, the tax represents 7.0% of final price for the 10% scenario, 12.9% for the 20% tax, and 18.0% for the 30% tax.

## Discussion

Using own, cross price elasticities and income elasticities from a Linear Approximation of the Almost Ideal Demand System (AIDS), we estimated the potential revenues on labeled food and beverages under three scenarios: 10%, 20%, and 30% of the final price without taxes. Results show that after accounting for a reduction in consumption after the taxes, economic and population growth, revenues for all groups could reach between 388,591.9 (10% tax) to 1,104,316.8 (30% tax) million Chilean pesos (about 528 million USD to 950 million USD). Ideally, if revenues are allocated to health or other welfare-related expenditures, the benefits of improved health will be combined with benefits from the government expenditures and will be very impactful on the welfare of Chileans.

Own and cross price elasticities are in line with other studies [47]. Labeled meat and fish had a lower elasticity, as expected, given that these are more basic foods. Price elasticities for beverages are also similar to a paper that estimated the impact of a combination of taxes on beverages and sweets and snacks on purchases and consumer welfare in Chile [47].

Our study has some limitations. We acknowledge that we do not have a complete food system as we used data from barcoded packaged food and beverage purchases. Basic food such as fruits, vegetables and other unpacked items are not collected. However, most of the bar coded labeled and unlabeled products are include in the data set. We also recognize that Kantar is representative of the urban population, price elasticities may be higher in rural areas [48, 49]. However, urban estimates are reasonable as Chile is mostly an urban country: 88% of the population lives in urban areas (population higher than 2,000 inhabitants) [50]. For revenue estimation, we used the total population to get national level figures.

It is important to note that from any taxation perspective, these added taxes would be considered regressive as the poor would pay a greater proportion of their income [4, 51]. However from a wider welfare perspective (that considers the impact on taxation on families' budget, but also private and social healthcare costs savings, as has been discussed with SSB taxes) the poor benefit greatly [5, 52]. They are most likely to have undiagnosed or poorly treated ultra-processed food related NCD [53].

The potential health benefits from reducing ultra-processed foods is large [15]. These foods are high in added sugars, added saturated fats, and sodium and usually have large proportions of unhealthy refined carbohydrates with excess calories. Reducing these foods in the Chilean diet will have significant impacts on obesity and NCD levels in Chile and produce profound improvements in health beyond the current set of regulatory actions. They will reinforce the impact of the warning labels on purchases and consumption of unhealthy ultraprocessed foods and beverages.

The advantage of using the approach followed in our study is that for any country that adopted a nutrient profile model that identified the unhealthiest foods and beverages such as the warning label system of Chile and other countries, they could follow a similar approach to align taxes or other policies with the warning labels to create mutually reinforcing fiscal and labelling set of laws. As shown here, the tax base for identifying ultra-processed foods and beverages, shown to have large impacts on health, is large and impactful on food purchases, and provides significant financial resources for a country. At the same time as in Chile, this same approach can be used for many other policies from identifying with warning labels those foods

and those to be banned from schools and other public facilities and for banning marketing of such products. It is more impactful if fiscal policies designed to promote healthy eating align with other policies like warning label or school ban policies to reinforce each other [14].

If Chile moves ahead and adopted this fiscal package, it will enhance significantly the health benefits of the current set of regulations aimed at obesity and NCD prevention. Furthermore, the revenues from these taxes can be used to enhance health and welfare in Chile. Their biggest benefit in health terms will be on the lower income Chileans.

## Supporting information

**S1 Table. Own, cross price elasticities and income elasticities for food and beverages 2015–2017 (QUAIDS model).**
(DOCX)

**S2 Table. Parameters for revenue estimation.**
(DOCX)

## Acknowledgments

We wish to thank Ariel Adams for administrative help. Donna Miles for programming support, and our Chilean collaborators Camila Corvalan and Marcela Reyes, University of Chile for support.

## Author Contributions

**Conceptualization:** M. Arantxa Colchero, Barry M. Popkin.

**Formal analysis:** M. Arantxa Colchero.

**Methodology:** Guillermo Paraje.

**Writing – original draft:** M. Arantxa Colchero, Barry M. Popkin.

**Writing – review & editing:** M. Arantxa Colchero, Guillermo Paraje, Barry M. Popkin.

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
