## [Decision Letter · Decision Letter 0]

6 Apr 2021

PONE-D-20-34987

The impacts on food purchases and tax revenues of a tax based on Chile’s nutrient profiling model

PLOS ONE

Dear Dr. Popkin,

Thank you for submitting your manuscript to PLOS ONE. After careful consideration, we feel that it has merit but does not fully meet PLOS ONE’s publication criteria as it currently stands. Therefore, we invite you to submit a revised version of the manuscript that addresses the points raised during the review process.

We look forward to receiving your revised manuscript.

Kind regards,

Maya K. Vadiveloo

Academic Editor

PLOS ONE

Journal Requirements:

Please provide additional details regarding participant consent. In the ethics statement in the Methods and online submission information, please ensure that you have specified (1) whether consent was informed and (2) what type you obtained (for instance, written or verbal, and if verbal, how it was documented and witnessed). If your study included minors, state whether you obtained consent from parents or guardians. If the need for consent was waived by the ethics committee, please include this information.

In your Methods section, we note the statement "We obtained nutrition facts panel (NFP) data from product photographs collected by a team of Chilean nutrition research assistants in stores during the first quarters of 2015, 2016, and 2017.". Please clarify how this dataset was accessed and how other researchers might access it."

In your Data Availability statement, you have not specified where the minimal data set underlying the results described in your manuscript can be found. PLOS defines a study's minimal data set as the underlying data used to reach the conclusions drawn in the manuscript and any additional data required to replicate the reported study findings in their entirety. All PLOS journals require that the minimal data set be made fully available. For more information about our data policy, please see http://journals.plos.org/plosone/s/data-availability.

Please include captions for your Supporting Information files at the end of your manuscript, and update any in-text citations to match accordingly. Please see our Supporting Information guidelines for more information: http://journals.plos.org/plosone/s/supporting-information.

Thank you for including your ethics statement: 'This research was reviewed by both Institutional review boards at the University of North Carolina and the National Institute of Public Health in Mexico.'      

6a.Please amend your current ethics statement to confirm that your named institutional review boards specifically approved this study.

6b. Once you have amended this/these statement(s) in the Methods section of the manuscript, please add the same text to the “Ethics Statement” field of the submission form (via “Edit Submission”).

Additional Editor Comments :

Thank you for submitting this interesting manuscript to PLOS One. Based on the thoughtful reviewer comments, I would be interested in sending out a substantively revised version of the manuscript for re-review. Notably, clarity regarding various terms, data sources, and methodological decisions would strengthen the manuscript.

Reviewers' comments:

Reviewer's Responses to Questions

**Comments to the Author**

1. Is the manuscript technically sound, and do the data support the conclusions?

Reviewer #1: Partly

Reviewer #2: Yes

2. Has the statistical analysis been performed appropriately and rigorously? 

Reviewer #1: I Don't Know

Reviewer #2: Yes

3. Have the authors made all data underlying the findings in their manuscript fully available?

Reviewer #1: No

Reviewer #2: Yes

4. Is the manuscript presented in an intelligible fashion and written in standard English?

Reviewer #1: No

Reviewer #2: Yes

5. Review Comments to the Author

Reviewer #1: This study addresses a relevant and important research question. However, the manuscript lacks structure, and the writing is sometimes confusing. Therefore, it is not clear to the reader how the study was carried out and it is hard to judge the quality of the research.

Abstract

• The background section is hard to understand. What is meant by “This study builds off a system of identification of unhealthy ultraprocessed foods that can be used by many countries with similar laws and the half dozen proposing the same system now”? Please rephrase this whole section, clearly stating what the objective of the study is.

• What is meant by “own”and “almost ideal” in the sentence “We derived own and cross price elasticities using the linear approximation of the almost ideal demand system for six groups of food and beverages”? Please rephrase.

• What is meant by labelled and unlabelled? It is not clear to readers who have not yet read the paper

• 1,299 million USD is 1.3 billion USD

Introduction

• Please include a definition of ultra-processed foods

• Line 71: What do you mean by “most important” NCDs

• Line 74-76. In the RCT you refer to, a diet of ultra-processed foods led to weight gain but other NCDs were not studied. Please rephrase and remove the words “most important” and “quite large”.

• Line 85: Highest sales where, in Latin America or in the world? Please specify.

• Line 86: Please change the word focuses throughout the manuscript as it is not correct to say that a tax focuses on certain foods. You can say that it is a tax on ultra-processed foods, or a tax that targets ultra-processed foods

• Line 88-89: it is not clear what the objective of your study is. Please clearly state the objective at the end of the introduction. For example, “We sought to determine….”

Methods

• What is Kantar WorldPanel Chile?

• Line 98: What is meant by “With replacement”

• What is Mintel Latin America?

• Line 129: “Most foods and beverages”, can you provide the %?

• Line 131-133: Do you mean foods carrying a FOP warning label, when you write labelled products? Please clearly state so in order to avoid confusing the reader

• Statistical analyses, Estimation of price elasticities: This section is a bit confusing and it is not clear to me what you did. Can you explain the analyses in a simpler and more straightforward way?

• Line 155-156: It is not clear to me what Fisher Price Index is and I am not familiar with the Laspeyres Price Index and the Paasche Price Index. Please explain further as many readers may not know what these indexes are.

• Line 168: I assume you mean tax revenues, please clarify in the text

• Lines 170-171: “Although SSB already have a tax, we applied the scenarios as if no taxes existed as the other food groups.” Please motivate this decision

Results

• Table 2: I am unfamiliar with your method of analysis and do not understand what table 2 shows or how you arrived at these results

Discussion

• Lines 239-241: please provide a reference

• Please include a paragraph describing the strengths and limitations of your study

Reviewer #2: This paper builds a demand system using Chilean scanner data to simulate the impact of a sin tax. The use of the Chile home scanner data are quite interesting. I also commend the authors for linking the data to nutrition data, which is some impressive work. Overall I think this study has potential to contribute to the literature meaningful. My comments are not substantial, with many of them clarification issues.

1. The paper talks about package labeling but the transition to tax is abrupt. Is the government actually considering a tax? What is the phase three Chilean food and beverage cutoffs? Some more background on this would clarify.

2. What is the degree of imputation for the data?

3. The paper assumes a 100% tax pass through. Is this (consumers bear the full tax burden) consistent with the literature? Chilean might be a special case if a value added tax is used, compared with the U.S., which uses either an excise unit tax or sales tax added at the register.

4. The authors state "non-purchases of any food or beverage within each group is very small, the percent of purchases greater than zero is more than 95% for all groups". This will make the demand modeling complicated because some of the budget shares will be zero. One complicated way to handle this is to estimate a demand system in the spirit of a tobit model allowing for zero budget share. In your case, zero purchases are small portion of the data, maybe a sensitivity check by dropping all the zero purchase data might help address the issue to certain degree.

5. Need clarification on the restrictions on the parameters, i.e., symmetry and homogeneity.

6. The income elasticities reported in the table should be labeled as expenditure elasticity since these are conditional on the food expenditure.

6. PLOS authors have the option to publish the peer review history of their article (what does this mean?). If published, this will include your full peer review and any attached files.

Reviewer #1: No

Reviewer #2: **Yes: **Yuqing Zheng

---

## [Decision Letter · Decision Letter 1]

16 Nov 2021

The impacts on food purchases and tax revenues of a tax based on Chile’s nutrient profiling model

PONE-D-20-34987R1

Dear Dr. Popkin,

We’re pleased to inform you that your manuscript has been judged scientifically suitable for publication and will be formally accepted for publication once it meets all outstanding technical requirements.

Kind regards,

Maya K. Vadiveloo

Academic Editor

PLOS ONE

Additional Editor Comments (optional):

Thank you for thoughtfully addressing both reviewers' comments. Please take note of the small editorial suggestions from both reviewers as you prepare your final manuscript.

Reviewers' comments:

Reviewer's Responses to Questions

**Comments to the Author**

1. If the authors have adequately addressed your comments raised in a previous round of review and you feel that this manuscript is now acceptable for publication, you may indicate that here to bypass the “Comments to the Author” section, enter your conflict of interest statement in the “Confidential to Editor” section, and submit your "Accept" recommendation.

Reviewer #1: All comments have been addressed

Reviewer #2: All comments have been addressed

2. Is the manuscript technically sound, and do the data support the conclusions?

Reviewer #1: Yes

Reviewer #2: Yes

3. Has the statistical analysis been performed appropriately and rigorously? 

Reviewer #1: I Don't Know

Reviewer #2: Yes

4. Have the authors made all data underlying the findings in their manuscript fully available?

Reviewer #1: Yes

Reviewer #2: Yes

5. Is the manuscript presented in an intelligible fashion and written in standard English?

Reviewer #1: Yes

Reviewer #2: Yes

6. Review Comments to the Author

Reviewer #1: Thank you for addressing my comments so thoroughly. Please proof read the article carefully as there are some typos in the edited text.

Reviewer #2: I think overall the authors addressed my comments successfully. I have two minor remaining suggestions: first, the authors might want to cite a few studies on page 11 to support their argument that "there is evidence from different counties that taxes fully passed to consumers". Second, as the the "The AIDS and QUAIDS models account for households with 177 zero purchases. The model takes average unit values as proxy for prices for all households 178 (purchasing or goods or not) and considers households with a zero-budget share (39)". I still have reservation on this point since I have not seen an empirical AIDS model paper that claims the normal AIDS applies to zero budget share. Otherwise, there will be no use for the development of a censored demand system.

7. PLOS authors have the option to publish the peer review history of their article (what does this mean?). If published, this will include your full peer review and any attached files.

Reviewer #1: No

Reviewer #2: **Yes: **Yuqing Zheng

---

## [Editor Report · Acceptance letter]

22 Nov 2021

PONE-D-20-34987R1 

The impacts on food purchases and tax revenues of a tax based on Chile’s nutrient profiling model 

Dear Dr. Popkin:

I'm pleased to inform you that your manuscript has been deemed suitable for publication in PLOS ONE. Congratulations! Your manuscript is now with our production department. 

Kind regards, 

on behalf of

Dr. Maya K. Vadiveloo 

Academic Editor

PLOS ONE